# Acoustic Based Fire Event Detection System in Underground Utility Tunnels

**Byung-Jin Lee** **, Mi-Suk Lee and Woo-Sug Jung** *

Electronics and Telecommunications Research Institute, Daejeon 34129, Republic of Korea;
byungjin.lee@etri.re.kr (B.-J.L.); lms@etri.re.kr (M.-S.L.)
* Correspondence: wsjung@etri.re.kr; Tel.:+82-10-2940-3441

**Abstract:** Underground utility tunnels (UUTs) are convenient for the integrated management of various infrastructure facilities. They ensure effective control of underground facilities and reduce occupied space. However, aging UUTs require effective management and preventive measures for fire safety. The fundamental problems in operating UUTs are the frequent occurrence of mold, corrosion, and damage caused to finishing materials owing to inadequate waterproofing, dehumidification, and ventilation facilities, which result in corrosion-related electrical leakage in wiring and cables. To prevent this, an abnormal sound detection technology is developed in this study based on acoustic sensing. An acoustic sensor is used to detect electric sparks in the moldy environments of UUTs using a system to collect and analyze the sound generated in the UUTs. We targeted the sound that had the highest impact on detecting electric sparks and performed U-Net-based noise reduction and two-dimensional convolutional neural network-based abnormal sound detection. A mock experiment was conducted to verify the performance of the proposed model. The results indicated that local and spatial features could capture the internal characteristics of both abnormal and normal sounds. The superior performance of the proposed model verified that the local and spatial features of electric sparks are crucial for detecting abnormal sounds.

**Keywords:** anomaly detection; acoustic sensing; deep learning; underground utility tunnel



## 1. Introduction

Underground utility tunnels (UUTs) are important infrastructure facilities required for modern urban life. The infrastructure typically accommodates two or more types of underground facilities, including electricity, gas, water supply, communication, and sewage facilities. UUTs overcome the topographical constraints of urban areas by using underground spaces. They are essential infrastructure facilities established for daily applications, such as communication, power, heating, and cooling amenities [1]. UUTs are known for easy maintenance, effective control of underground facilities, and reduced occupied space. Despite the high initial construction investment, these facilities continue to be installed in new cities as they are economically efficient in the long term [2]. In Korea, UUTs are installed in 30 regions to provide integrated services, such as electricity, water, and communication. Among them, facilities that have been in use for more than 30 and 20 years account for 25 and 43%, respectively. Numerous individual facilities are expected to age, and the management of aging UUTs is crucial because of increasing fire safety-related disasters [3].

Therefore, to ensure the safety and efficient maintenance of underground facilities that involve frequent accidents and warrant prompt response to disasters, smart technology-based systems have been proposed. A system [4] has been proposed to check the state of UUTs for abnormalities in facilities and installed equipment based on the data collected from closed-circuit television (CCTV) and various sensors, such as accelerometers and optical sensors. With the development of graphics processing units (GPUs) and the presentation

of various image classification methods based on convolutional neural networks (CNNs), deep learning techniques applied to artificial neural networks are rapidly advancing, exhibiting outstanding performance in the field of image recognition [5]. The integration of novel detection technologies and data-based algorithms is essential for constructing reliable and intelligent fire detection systems.

However, the fundamental operational problems of UUTs are the frequent occurrence of condensation, corrosion, and other issues owing to inadequate waterproofing, dehumidification, and ventilation facilities, which adversely affect the durability of underground structures and internal facilities. As depicted in Figure 1, condensation occurring inside utility tunnels damages the finishing materials owing to rust or various types of fungi, causing electrical leakage because of corroded cables [6–9]. The installation of various smart devices increases the number of cables to be managed, whose insulation may be damaged when retained in a condensed environment for long periods. Short circuits may occur if the cable with damaged insulation comes into contact with an aged cable, leading to a fire. In most fire incidents that have occurred thus far in domestic and international underground facilities, cable short circuits and thermal contact caused by combustible cables have been the primary causes of fire [10,11]. Although forced ventilation is executed by running fans to prevent condensation inside the UUTs, the aging of the facility and insufficient fan capacity often result in inadequate ventilation. Additionally, on rainy days during summer, the ventilation system often shuts down, leading to poor ventilation within underground structures [12].

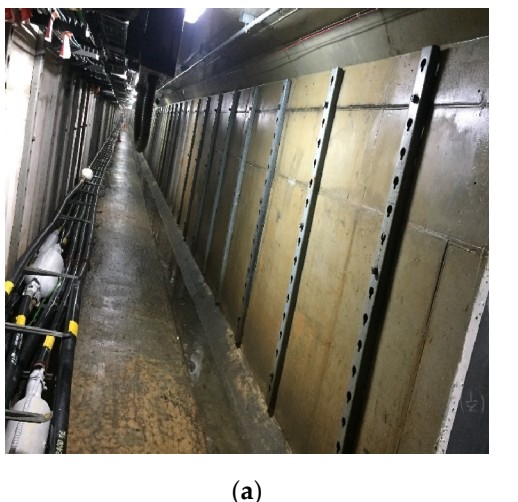

(**a**)

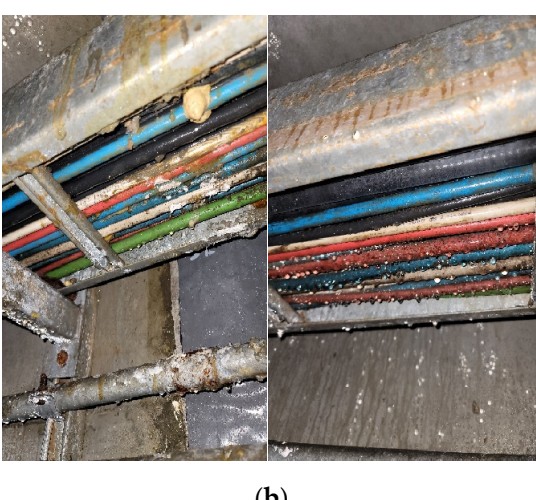

(**b**)

**Figure 1.** Condensation environment in underground utility tunnels (UUTs): (**a**) Condensation in UUTs; (**b**) Corrosion of cables and trays owing to condensation.

Conventional CCTV-based technology cannot cover an entire area because of blind spots, and existing spark detectors or infrared flame detectors are extremely large and expensive to be installed in UUTs. Therefore, an acoustic sensing-based anomaly detection system is developed in this study to prevent fires in the condensation environment of UUTs by detecting electric sparks. The proposed system with no visual limitations ensures relatively flexible installation and wide-range monitoring. Furthermore, the developed system is not a contact-type sensor, such as a temperature sensor; therefore, it can be installed in vulnerable areas where condensation occurs for a more efficient operation. It can be safely managed without interaction or interference with the equipment and devices installed inside the underground complex. As sound is affected by the surrounding noise, the proposed system measures and analyzes the sounds generated in the underground complex environment, uses a deep learning technique to improve the monitoring performance of detecting initial signs of overheating, such as electric sparks, and recommends a suitable methodology for training the datasets.

The remainder of this paper is structured as follows. Section 2 presents the methodology, architecture, and key components of the proposed anomaly detection system based on acoustic sensing. Section 3 describes the experimental settings, testing procedures, detailed experimental results, and evaluation of the performance of the system. Finally, the conclusions and suggestions for future work are summarized in Section 4.

## 2. Methodology

### 2.1. Acoustic Analysis of the Condensation Section of the UUTs

According to facility regulations, UUTs require natural and mechanical ventilation systems installed at regular intervals. Typically, UUTs experience a summer condensation problem in certain areas, as indicated in Figure 2. This condensation is caused by insufficient resistance of exterior materials to vapor penetration, the use of materials with high humidity, inadequate insulation in the construction owing to damaged or poor adhesion of insulation materials, and poor sealing of gaps. Therefore, high humidity is generated in the air inside the structure, and condensation occurs when the surface temperature of the interior walls or surfaces of the pipes reaches the dew point. To reduce condensation in the condensation zone, a mechanical ventilation system was operated; however, this generated noise from the ventilation fan. External noise, such as traffic noise, honking of vehicles, and wind noise, also entered the natural ventilation systems. Additionally, a manhole existed in the connecting passage of the underground building, and noise was generated from the manhole cover owing to the continuous impact of vehicle traffic. Other sources of noise within UUTs included the sound of water falling into the sump pit owing to the inflow of water or condensation through the manhole, as well as the operating sounds of devices, such as CCTV or electrical equipment installed in the surrounding area.

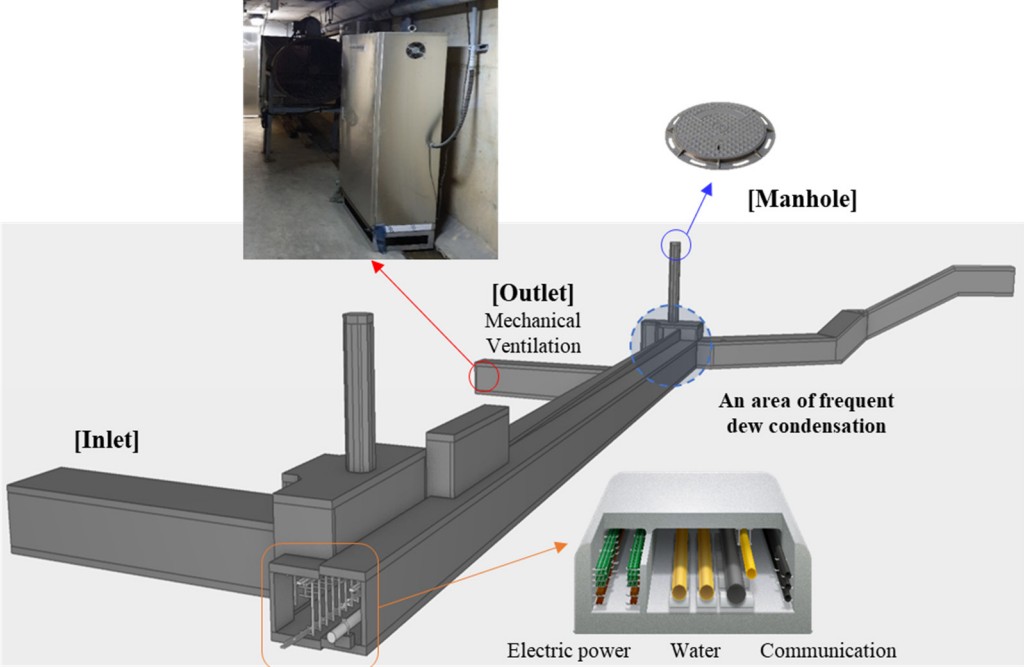

**Figure 2.** Environment around the condensation section of a UUT.

### 2.2. System Architecture

In this study, an anomaly detection system based on acoustic sensing is proposed to prevent potential fires in UUTs. As indicated in Figure 3, the primary components of the system included acoustic sensors for collecting acoustic data and an anomalous sound detection (ASD) server. In the case of condensation in UUTs, acoustic sensors should ideally be installed in or near the corridors of UUTs where the hazard is located. The acoustic sensor [13] is designed to suit the environment of the UUTs. To apply the sensor to actual

UUTs, it was modified to allow power and data transfer through a single Ethernet cable using a Power over Ethernet (PoE) switch as Figure 4. The PoE is substantially safer than a typical alternating current (AC) power supply that always supplies power to the outlet. Moreover, as PoE uses low voltage, difficult requirements such as separate wire conduits or distribution panels need not be considered. The i2s (Inter IC Sound) interface on the Raspberry Pi was configured to receive PCM (Pulse Code Modulation) data directly from the MEMS microphone using PDM mode. The sound acquisition H/W device made based on this configuration is IP (Ingress Protection rating) 67 waterproof to prevent abnormalities caused by condensation during sound acquisition. For the microcontroller unit, we used a Raspberry Pi ZERO and connected it to the microphone sensor to measure the data. The acoustic sensor module consists of Common Audio Driver in the kernel driver area and Pulse Density Modulation (PDM) driver, and the application consists of PCM capture, Advanced Audio Coding (AAC) encoder, real-time transport protocol (RTP) packetizer, User Datagram Protocol (UDP) sender, etc. The PDM Driver is written by modifying the I2S module of the linux kernel, is connected to the digital microphone versper VM3000 and the PDM interface, and is responsible for receiving acoustic data with a sampling rate of 48,000 and 24 bits. The sound data were compressed using an audio codec (AAC-LC) and transmitted to an ASD server through a transmission router using the RTP.

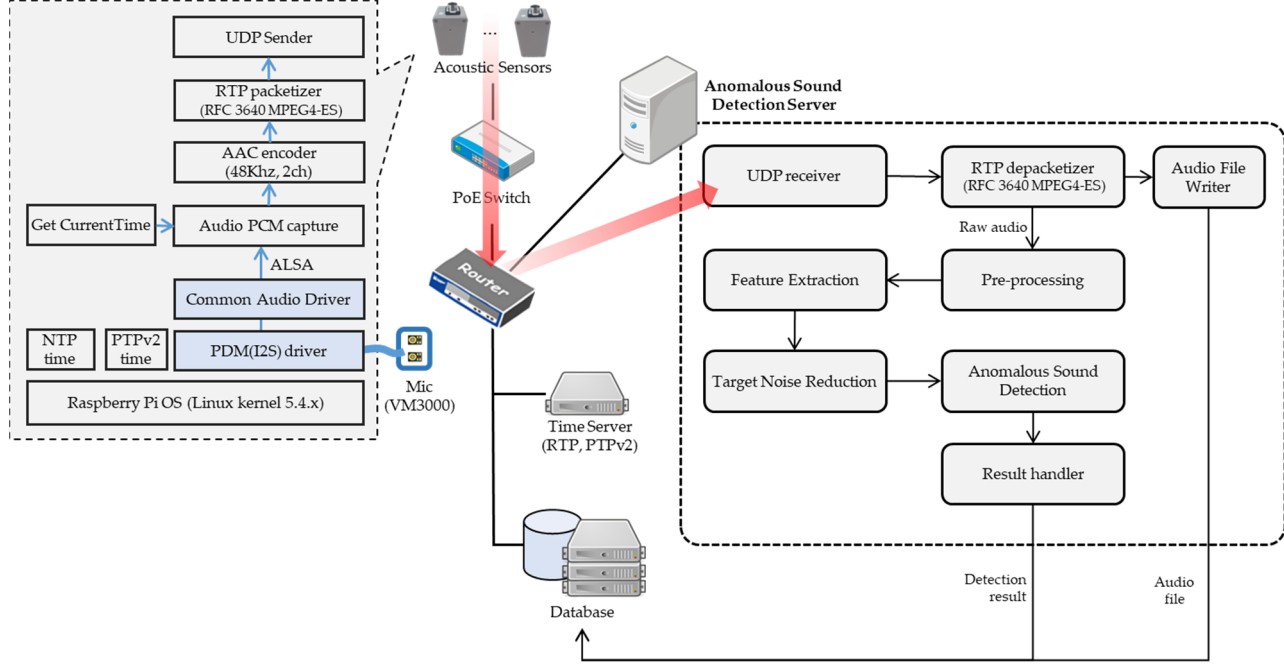

**Figure 3.** Overall diagram of the proposed acoustic sensing-based anomaly detection system.

The UDP Receiver in the anomalous sound detection server receives RTP packets sent from the acoustic sensor. RTP depacketizer parses the RTP packets to obtain audio elementary stream (ES), synchronization source (SSRC), and time information. Audio files are stored in the database through Audio File Writer. The PCM data were subjected to pre-processing, feature extraction, and target noise reduction (TNR). Subsequently, the ASD deep learning model tested whether the sound signal indicated an anomaly and transmitted the result and corresponding sound data to the database.

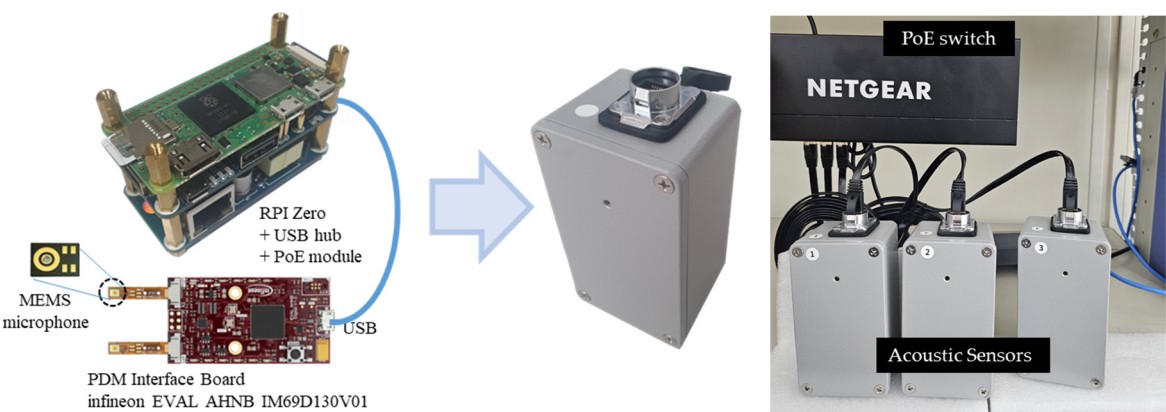

**Figure 4.** The modified acoustic sensor.

*2.3. Data Acquisition and Pre-Processing*

UUTs are restricted in terms of access, rendering it difficult to acquire acoustic data during fire or disaster situations. Moreover, electric sparks cannot be generated in underground cavities owing to the risk of fire. As electric spark sounds are not commonly experienced in daily life, artificially created sound datasets are commonly used. Therefore, we selected the sound that was most similar to the sound of an actual spark that occurred at the location where condensation was generated in the UUTs.

Since the voltage of the UUT management cable is 110 V/220 V, the sound of electric sparks corresponding to that voltage was collected from an open-source sound database. The collected sound data and a sound file similar to the spark generated by the actual UUT were played through a speaker at the UUT location where the condensation occurred.

Additionally, based on the sound data collection device and program reported in [13], the sound was collected from various locations in the UUTs, focusing on areas where condensation occurred. Figure 5 illustrates a representative sound generated inside the UUTs, with the spectrogram captured at 1 s intervals. Data were collected at the same location for 10 sessions from December 2021 to August 2022. We collected the sounds of noise and electric sparks in UUTs during winter and spring and the sound of the ventilation fan during the vulnerable period of summer. After data collection, inspection, labeling, and labeling inspection, 564 GB of raw data were secured. The analysis of the sound collected inside the UUTs indicated that multiple sounds were reflected owing to their characteristics, and the use of an omnidirectional VM3000 MEMS microphone resulted in the creation of reverberations. An electric spark sound occurred in less than a second with the energy distribution ranging between 8 and 18 kHz. The sound of impact on the manhole exhibited a signal length of approximately 0.5 to 2 s owing to the speed differences of passing vehicles and consecutive passing of two or more vehicles, with energy occurring up to approximately 19 kHz. Notably, the sound was strongly concentrated below 3 kHz.

The most significant noise in the underground spaces was the sound of the ventilation fan. The energy was generated up to the 24 kHz frequency band, with a strong concentration below 18 kHz. This overlapped with the frequency band of the electric spark sounds.

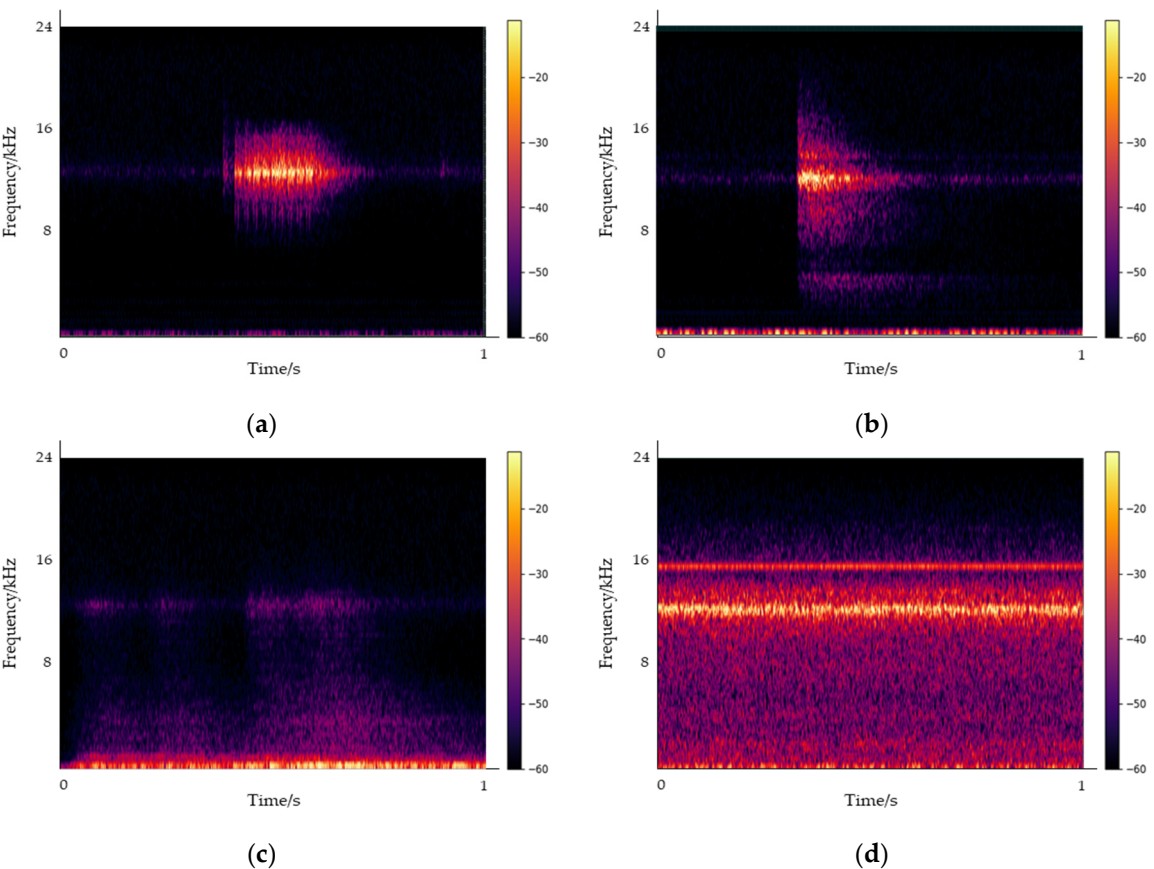

**Figure 5.** Representative sound measurement data inside the UUTs: (**a**) Electric spark; (**b**) Impact sound from the manhole caused by a passing car; (**c**) Sound of water dropping into the sump; (**d**) Sound of the fan operated in the ventilation duct.

The collected PCM data were transformed into magnitude spectrograms every 1 s using short-time Fourier transform (STFT). Therefore, the pre-processed one-dimensional (1D) fast Fourier transform (FFT) numerical data contained 48,000 points.

$$X_n\left(e^{jw}\right) = \sum_{l=-\infty}^{\infty} x(l)w(n-l)e^{j\omega n}, \tag{1}$$

where $w(n)$ denotes the Hamming window function, and $X_n\left(e^{jw}\right)$ indicates the function of $\omega$ and $n$. If $\omega = \frac{2\pi k}{N}, 0 \leq k \leq N-1$, N represents the number of FFTs. Here, each signal was subjected to FFT, and the corresponding STFT was obtained as

$$\begin{aligned} X_N(k) &= X_N e^{(2\pi kj/N)} \\ &= \sum_{l=-\infty}^{\infty} x(l)w(n-l)e^{-2\pi kj/N}. \end{aligned} \tag{2}$$

The short-time power spectrum $P_n\left(e^{j\omega}\right)$ was calculated as

$$P_n\left(e^{j\omega}\right) = X_N\left(e^{j\omega}\right)X_N\left(e^{j\omega}\right) = \left|X_N\left(e^{j\omega}\right)\right|^2, \tag{3}$$

$$P_n\left(e^{j\omega}\right) = \sum_{l=-\infty}^{\infty} R_n(k)e^{j\omega k}, \tag{4}$$

$$R_n(k) = \sum_{m=-\infty}^{\infty} x(l)w(n-l)x(l+m)w(n-l-m), \tag{5}$$

where $R_n(k)$ denotes the short-time autocorrelation of $x(n)$; $P_n\left(e^{j\omega}\right)$ indicates the Fourier transform of $R_n(k)$, and n and ω represent the horizontal and vertical coordinates, respec-

tively. STFT with a window size of 512 was performed using a Hamming window and 63% overlap to maintain the time resolution less than 0.01 s. As the sound of electric sparks was short, the time resolution had to be increased. Additionally, we set a minimum frequency resolution of approximately 100 Hz to satisfy the requirements of the high frequency range of the sound. Consequently, the 1D time-domain data of 1 s were converted into 1D STFT data using 256 times FFT. The size of each STFT dataset with 48,000 points was reduced to create a dataset of size 256 × 256. This was used as a feature of the deep learning model. In the spectrogram, the range of variation of the harmonics was (0, 24,000) Hz. The Butterworth bandpass filter was designed to pass the signal through the band at 4000 and 20,000 Hz. This was a combination of high-pass and low-pass filters with cutoff frequencies of 4000 and 20,000 Hz, respectively. This range was filtered because it comprised most of the primary information of the spark signal.

### 2.4. Deep Learning Models

As illustrated in Figure 6, two different deep learning models were considered, each with two different training datasets. Typically, the systems operating in the spectrogram domain use the mixed-signal phase when restoring the signal in the time domain. However, errors may occur in the estimated signal because of using the mixed-signal phase. To overcome this drawback, the desired signal was directly estimated in the frequency domain without being transformed into the time domain.

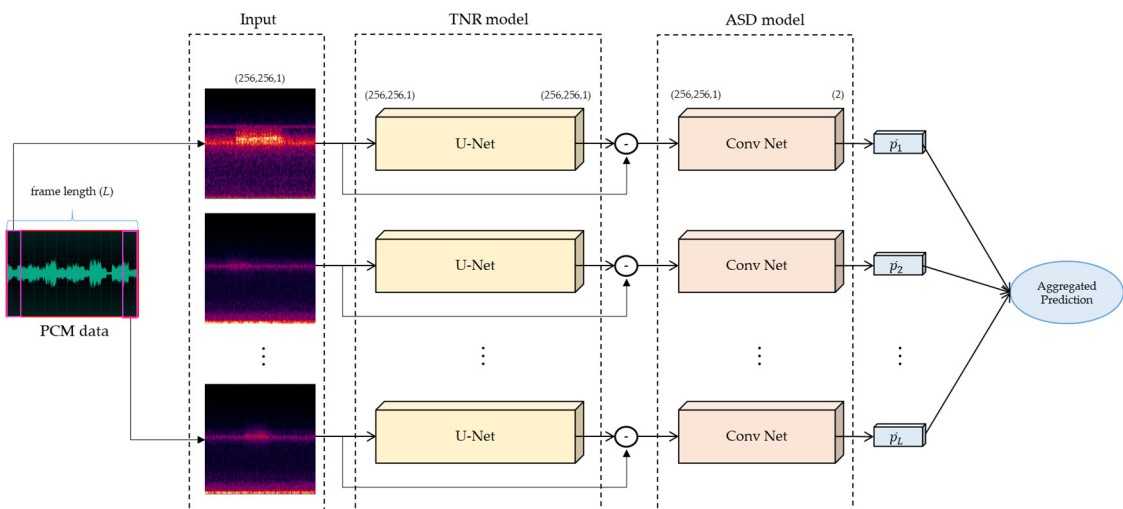

**Figure 6.** Structure of the proposed U-Net + convolutional neural network (CNN) anomalous sound detection (ASD) models.

### 2.4.1. U-Net-Based TNR Model

The U-Net model was used for training, which is a deep convolutional autoencoder originally developed for image segmentation in the medical field [14]. U-Net utilizes features with both overall contextual information and precise localization; in this study, U-Net was adapted to remove noise spectra. Figure 7 illustrates the network architecture used for training. The network input was the mixing spectrum of sparks and ventilation fans with dimensions of 256 × 256. Both the input and output matrices were globally scaled and mapped to a normalized distribution between −1 and 1. The encoder comprised 10 convolutional layers with the leaky rectified linear unit (LeakyReLU), max pooling, and dropout layers. The decoder was a symmetrically expanding path with skipped connections. The final activation layer was a hyperbolic tangent (tanh) with an output distribution between −1 and 1. The model was compiled using the Adam optimizer, and the loss functions were used as a trade-off between $L_{L1}\left(y, y_{TNR}^{p}\right) = \sum_{i}\left|y_i - y_{TNR,i}^{p}\right|$

and $L_{\text{SquaredL2}} = \sum_i \left| y_i - y_{TNR,i}^p \right|^2$. Furthermore, the Huber loss and log-cosh loss were compared and analyzed. Here, $y_i$ denotes the $i$th entry of $y$; $y$ indicates the actual value, and $y_{TNR}^p$ represents the value predicted by the TNR model. Owing to the strong noise characteristics of the ventilation fan, several outliers existed in the signal. Therefore, a robust loss function resistant to outliers was used to compensate for this.

$$L_{huber}\left(y, y_{TNR}^p\right) = \begin{cases} \frac{1}{2}\left(y, y_{TNR}^p\right)^2, for \left|y - y_{TNR}^p\right| \leq 1 \\ \left|y - y_{TNR}^p\right| - \frac{1}{2}, for \left|y - y_{TNR}^p\right| > 1 \end{cases} \tag{6}$$

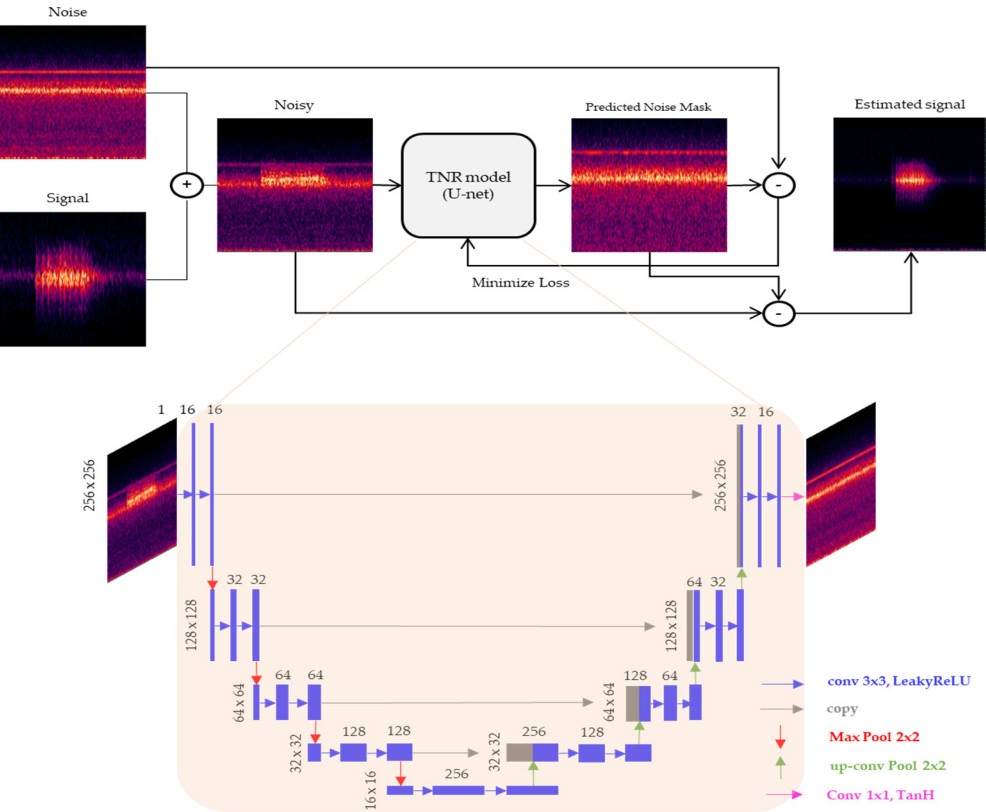

**Figure 7.** U-Net architecture for target noise reduction.

The Huber loss function addressed the drawbacks of non-differentiable L1 loss by applying L2 loss when the value was less than 1 and L1 loss when the error was greater than 1, thus combining the strengths of the two types of loss functions [15]. The use of the Huber loss function during the training of the TNR model ensured rapid learning even when large errors occurred. Using the robust L1 loss at the beginning of the training process when the TNR model could not converge was effective. As the training progressed and the loss function reduced to a value less than one, stable learning was achieved by using the differentiable L2 loss.

$$L_{cosh}\left(y, y_{TNR}^p\right) = \sum_{i=1}^n log\left(\cosh\left(y_{i,TNR}^p - y_i\right)\right). \tag{7}$$

The log-cosh loss function is another loss function applied to regression problems that are smoother than the mean squared error (MSE). This function combines the advantages of the MSE and mean absolute error (MAE), reduces the sensitivity to outliers, and improves the robustness of the neural network model to outliers.

A mask was generated to predict noise using this process, and the noise-free signal was estimated by directly subtracting the noisy input signal from the output of the model.

### 2.4.2. Two-Dimensional (2D) CNN-Based ASD Model

A 2D-CNN structure was used to detect anomalous sounds with magnitude spectrograms of size [256 × 256 × 1] as input. Figure 8 illustrates the detailed structure of the 2D CNN used for ASD. The Conv Net comprised four 2D convolutional layers, one global average pooling (GAP) layer, and two dense layers. The kernel size of the 2D convolutional layers was [3,3], with padding set to "same" and strides set to 1 only for Conv1 and 2 for Conv2–Conv4. Batch normalization was applied to each layer (Conv), and the exponential linear unit (ELU) was used as the activation function. After the fourth convolutional layer (Conv1–Conv4), the model was lightweight, and overfitting was prevented using GAP. In general, GAP is used for extracting features from each sample by repeatedly performing average pooling to reduce the sample size and extract representative values of [1,1] filter size from each sample. Therefore, a sample size of [32 × 32, 256] in Conv4 was represented using a [256] layer with 256 representative values. Subsequently, the model was passed through two dense layers, and the prediction probability was calculated using the SoftMax activation function.

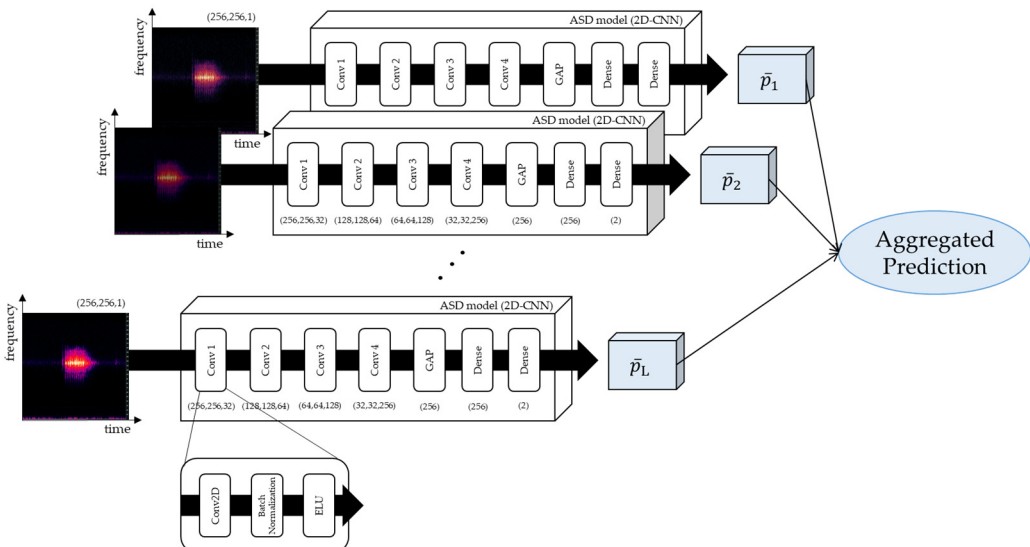

**Figure 8.** Architecture of the two-dimensional CNN for ASD.

Next, the fusion of predicted probabilities obtained from the ASD model based on the frame length was used for the analysis. The fusion result $P_f = (\overline{p}_1, \cdots \overline{p}_L)$ was obtained as follows:

$$\overline{p}_L = \frac{1}{L}\sum_{l=1}^{L} \overline{p}_{lc}, \, for\, 1 \le l \le L, \tag{8}$$

where $\overline{P}_L = (\overline{p}_{l1}, \cdots \overline{p}_{lC})$ denotes the predicted probability of 1D time-domain data of 1 s; $C$ indicates the category number ($C$ = 2), and $l^{th}$ was evaluated considering L frames. The aggregated predicted label $\hat{y}$ was determined as

$$\hat{y} = argmax(\overline{p}_1, \cdots \overline{p}_L). \tag{9}$$

## 3. Experiments and Results

### 3.1. Experimental Setup

The UUT for the experiment is composed of three rows as shown in Figure 2, and the receiving facilities are maintained and managed as telecommunications, water, and electricity. Natural ventilation facilities and forced ventilation facilities powered by electricity are installed around the vulnerable areas where actual condensation is occurring, and forced ventilation is always operated in summer when humidity is high, but it is not effective in preventing condensation due to insufficient ventilation capacity due to the aging of

ventilation fans [16]. Therefore, the management tray and cables are corroded as shown in Figure 1b. Although the area is being monitored by a fixed CCTV, it can only roughly observe the passage for external intrusion detection, and the cable tray is located in the CCTV blind spot, making it impossible to monitor.

The experiments were classified into two primary groups, namely, TNR and ASD experiments. We performed TNR to detect spark signals in the ventilation fan environment, with the ventilation fan noise set as the target noise. Noises were blended with sparks for noise reduction, with randomization of the noise level between 20 and 80%. As summarized in Table 1, 360,550 data samples were used for training and validation, and the time of each sample was 1 s. Of the total data, we used 75% (270,412 samples) as training data and 25% (90,138 samples) as test data. Based on the measured data, a clean spark was selected as the acoustic signal without mixing with other noises. The acoustic data used for ASD in this experiment were collected from actual UUTs; 543,692 data samples were used for training and validation, and the time of each sample was 1 s. The data were divided into 70% training data (380,584 samples) and approximately 30% test data (163,108 samples). Here, 10% of the training data was used for validation. We performed 10-fold cross-validation, considering the average of eight models of ten with the highest accuracy; the models with the highest and lowest accuracies were excluded. Adam optimization function was used during training, with a learning rate of 0.001 and a mini-batch size of 64. The number of epochs was set to 200; however, early termination was applied if the validation loss did not decrease after 20 epochs. Additionally, to verify the model in an actual environment, the ventilation fan in the UUT was operated, and the sound data of the ventilation fan + electric spark were collected (12,572 samples).

**Table 1.** Number of training and test datasets used for each model.

| TNR Model | | | ASD Model | | |
|---|---|---|---|---|---|
| Training data | Test data | Total | Training data | Test data | Total |
| 270,412 | 90,138 | 360,550 | 380,584 | 163,108 | 543,692 |

*3.2. Evaluation Metrics*

To evaluate the efficiency of the proposed model, the commonly used source-to-distortion ratio (*SDR*) [17] was considered as a training target and evaluation metric. This negative *SDR* has the advantage of preserving the scale and matching the mixture. The *SDR* indicates the difference between the true source s and estimated source ŝ in the time domain.

$$SDR(s, \hat{s}) = 10\log_{10} \frac{\|s\|^2}{\|s - \hat{s}\|^2}. \tag{10}$$

Furthermore, accuracy, precision, recall, and F1-score were considered as performance evaluation metrics. All the evaluation metrics were calculated using different attributes within the confusion matrix. In a typical confusion matrix, true positive (TP) and true negative (*TN*) denote the correctly predicted anomaly and normal instances, respectively, whereas false negative (*FN*) and false positive (FP) represent the incorrectly predicted instances of normality and anomaly, respectively. The evaluation metrics considered in this study can be summarized as follows [18]:

$$Accuracy = \frac{TP + TN}{TP + TN + FP + FN}, \tag{11}$$

$$Precision = \frac{TP}{TP + FP}, \tag{12}$$

$$Recall = \frac{TP}{TP + FN}, \tag{13}$$

$$F1\_Score = 2 \times \left( \frac{Precision \times Recall}{Precision + Recall} \right). \tag{14}$$

### 3.3. Results

Table 2 lists the average SDR improvements of the TNR model for the test dataset with different loss functions. Considering one mixture with SDR as an example, Figures 9 and 10 illustrate the normalized spectrograms of the mixture with signal-to-noise ratios (SNRs) of −10 and 0 dB, respectively; the figures indicate four estimated sources obtained via TNR with each loss.

**Table 2.** Average source-to-distortion ratio (SDR) improvement (dB) on the test dataset for the target noise reduction (TNR) model considering different loss models.

|  |  | SNR | | | | | |
|---|---|---|---|---|---|---|---|
|  |  | −20 dB | −10 dB | −5 dB | 0 dB | 5 dB | 10 dB |
| SDR | L1 loss | −15.93 | −5.42 | −0.15 | 4.1 | 7.31 | 10.08 |
|  | L2 loss | −17.1 | −3.09 | 1.23 | 4.89 | 8.04 | 10.98 |
|  | Huber loss | −19.74 | −8.34 | −0.15 | 5.28 | 8.39 | 11.07 |
|  | cosh loss | −12.93 | −2.81 | 1.24 | 4.87 | 8.01 | 10.41 |

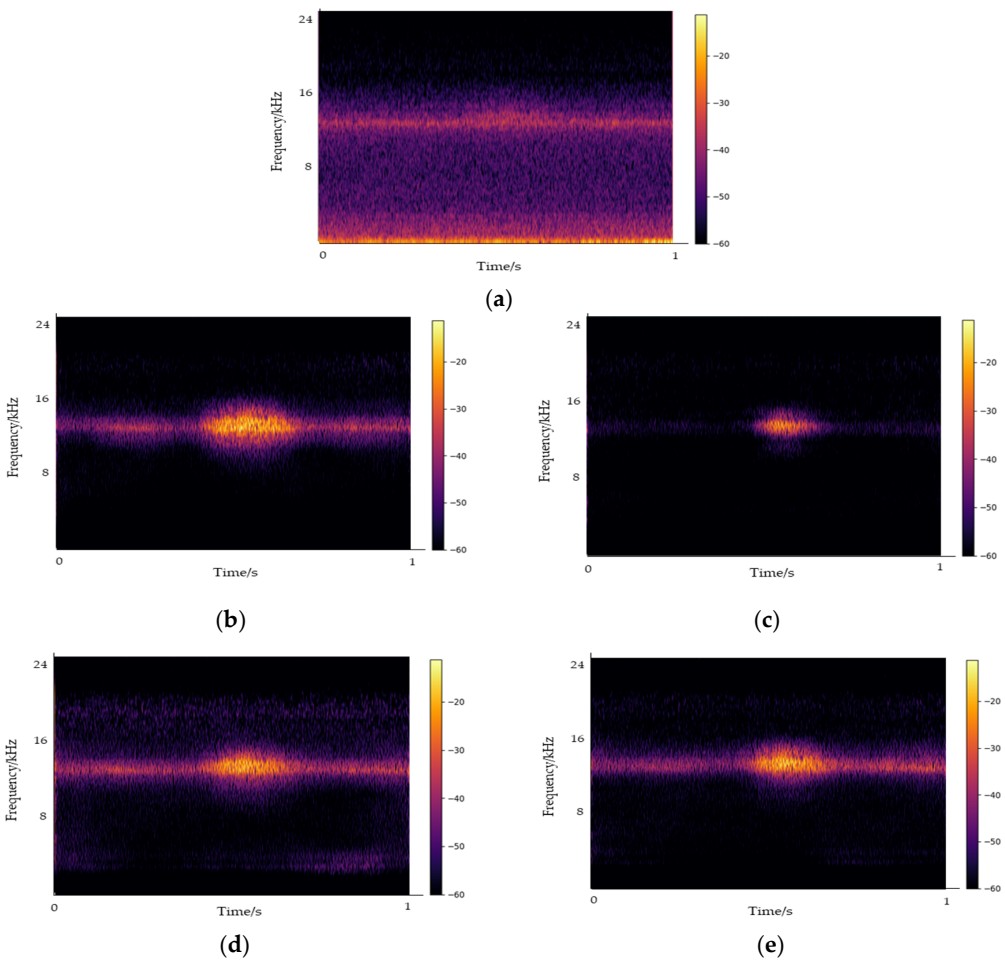

**Figure 9.** Short-time Fourier transform (STFT) analysis for a signal-to-noise ratio (SNR) of −10 dB: (**a**) mixture; (**b**) log-cosh; (**c**) Huber; (**d**) L1; (**e**) L2.

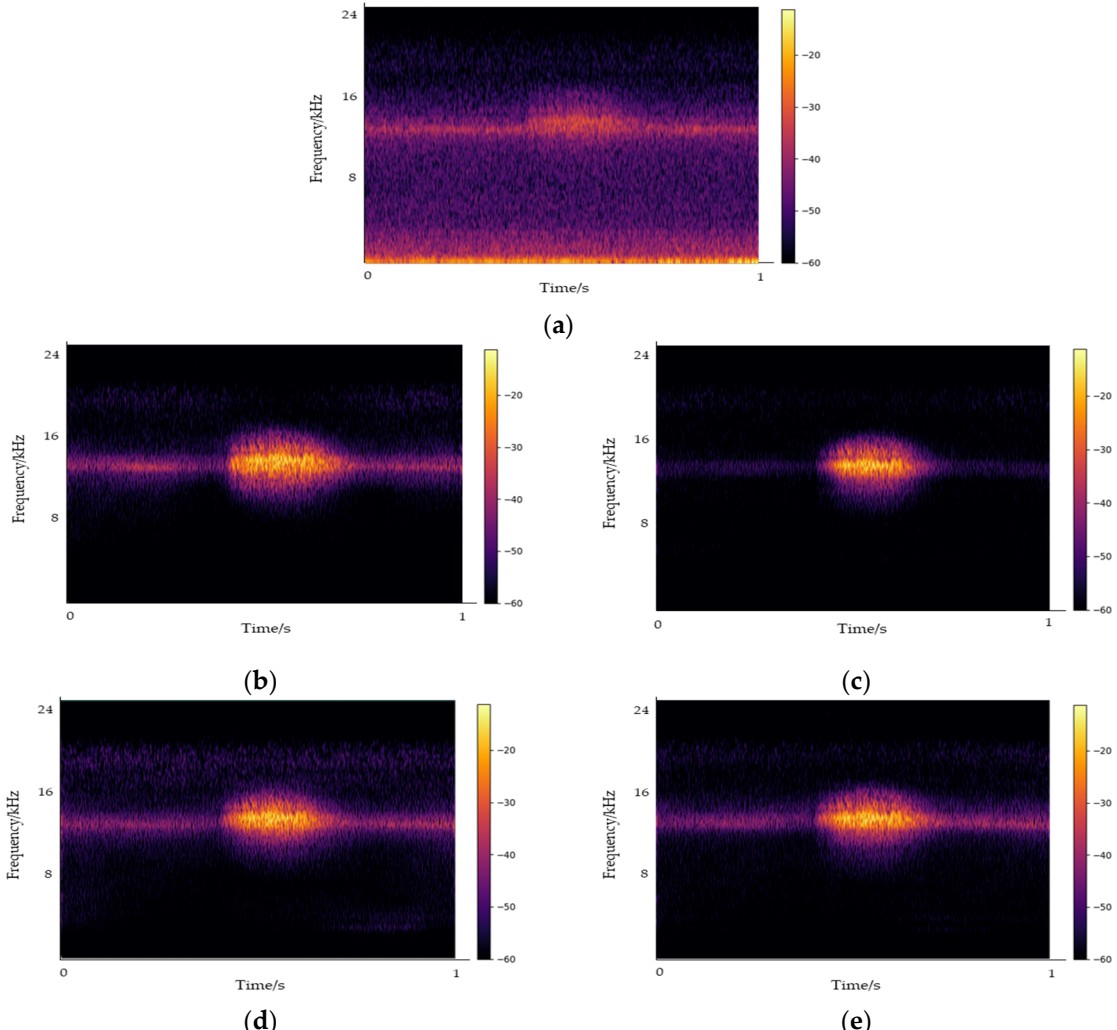

**Figure 10.** STFT analysis for an SNR of 0 dB: (**a**) mixture; (**b**) log-cosh; (**c**) Huber; (**d**) L1; (**e**) L2.

During the experiment, the best loss function was determined to be the log-cosh for SNRs below 0 dB, whereas the Huber loss was more effective for higher SNRs. Although the log-cosh estimated the spectrum of the spark relatively well even at low SNRs, residual noise remained. Therefore, the SDR of the log-cosh loss was lower than that of the Huber loss at an SNR of 0 dB, where the spark spectrum was clearly visible. For L1 loss, the performance was lower than that of log-cosh, whereas L2 loss exhibited a performance similar to that of log-cosh for 0 dB and higher SNRs. As log-cosh loss was less sensitive to outliers, its performance was relatively high in low SNRs; however, it was more sensitive to small errors. Thus, the prediction performance of L2 loss was relatively high for SNRs of 0 dB and higher. The Huber loss effectively suppressed residual noise at high SNRs, resulting in a higher SDR in comparison with that of the log-cosh. However, although certain strong spectral regions were estimated at low SNRs, the remaining portions could not be separated well from the mixture spectrum. This resulted in an SDR lower than that of the log-cosh. Therefore, log-cosh and L2 were advantageous for low SNRs in the UUT environment, and the L2 loss appeared to be suitable as a loss function for the TNR when compared in terms of the overall performance.

To evaluate ASD performance, we compared the three baseline models by adding them. In the anomaly detection problem in UUTs, recall and precision are more important than accuracy because samples with normal data are generally more than those with anomalous data. As the problem of data imbalance per class continues to occur in actual field situations, we evaluated the data based on performance evaluation metrics for anomalous

data. As indicated in Table 3, among the three models, the CNN exhibited the ability to extract the frequency characteristics of electric sparks with the highest performance. This was because local characteristics in the frequency domain of electric sparks were more prominent than temporal correlations in the collected data. Owing to the significant impact of misclassifying normal data as anomalous, the CNN with high recall was considered to exhibit better performance.

**Table 3.** Model performance evaluation of convolutional neural network (CNN), recurrent neural network (RNN), and long short-term memory (LSTM) for the anomalous sound detection (ASD) model.

| Model | Precision | Recall | F1-Score | Accuracy |
| --- | --- | --- | --- | --- |
| CNN | 0.9996 | 0.9998 | 0.9997 | 0.9998 |
| RNN | 0.9670 | 0.8888 | 0.9262 | 0.9504 |
| LSTM | 0.9877 | 0.9583 | 0.9865 | 0.9913 |

In this experiment, as anomaly detection in the frequency domain was performed using spectrograms, the performance of the ASD model had to be evaluated after being passed through the TNR model. Therefore, for actual environments, we evaluated the performance on the 12,572 (including 5518 anomalies) ventilation fan + electrical spark data points described in Section 3.1; as the loss function of the TNR model, we applied L2 derived from the above results. The results are presented below.

Figure 11 depicts the output of the TNR model on the actual ventilation fan + electric spark data collected from the UUTs. The data collected from the UUTs exhibited an SNR level ranging from 0 to 10 dB, and the TNR model output showed localized features of the electric sparks. Table 4 summarizes the results of detecting anomalous sounds by feeding the output of the TNR model into the ASD model. In most cases, TNR-CNN exhibited the best performance. The overall accuracy of classifying the two categories was 0.9631. The anomaly detection precision, recall, and F1-score were 0.8953, 0.9976, and 0.9436, respectively. This indicated that the local characteristics could capture the internal characteristics of both abnormal and normal sounds. The better performance of the TNR-CNN verified that the local characteristics of electric sparks were important for detecting the abnormal sounds. Additionally, the recurrent neural network (RNN) and long short-term memory (LSTM) exhibited low results because they considered only the information of past datasets and the sequential information of electric sparks, whereas the characteristics of the noise-suppressed data obtained through TNR were not reflected. In this experiment, the data collected by the acoustic sensors accurately detected abnormal sounds even in severely noisy environments, such as when the ventilation fans were operated. In other words, acoustic sensors can serve as an effective sensing technology for early fire detection caused by electric sparks.

**Table 4.** Evaluation of the TNR + ASD model.

| Model | Precision | Recall | F1-Score | Accuracy |
| --- | --- | --- | --- | --- |
| TNR + ASD (CNN) | 0.8953 | 0.9976 | 0.9436 | 0.9631 |
| TNR + ASD (RNN) | 0.5846 | 0.9134 | 0.7129 | 0.7934 |
| TNR + ASD (LSTM) | 0.9757 | 0.7357 | 0.8389 | 0.8355 |

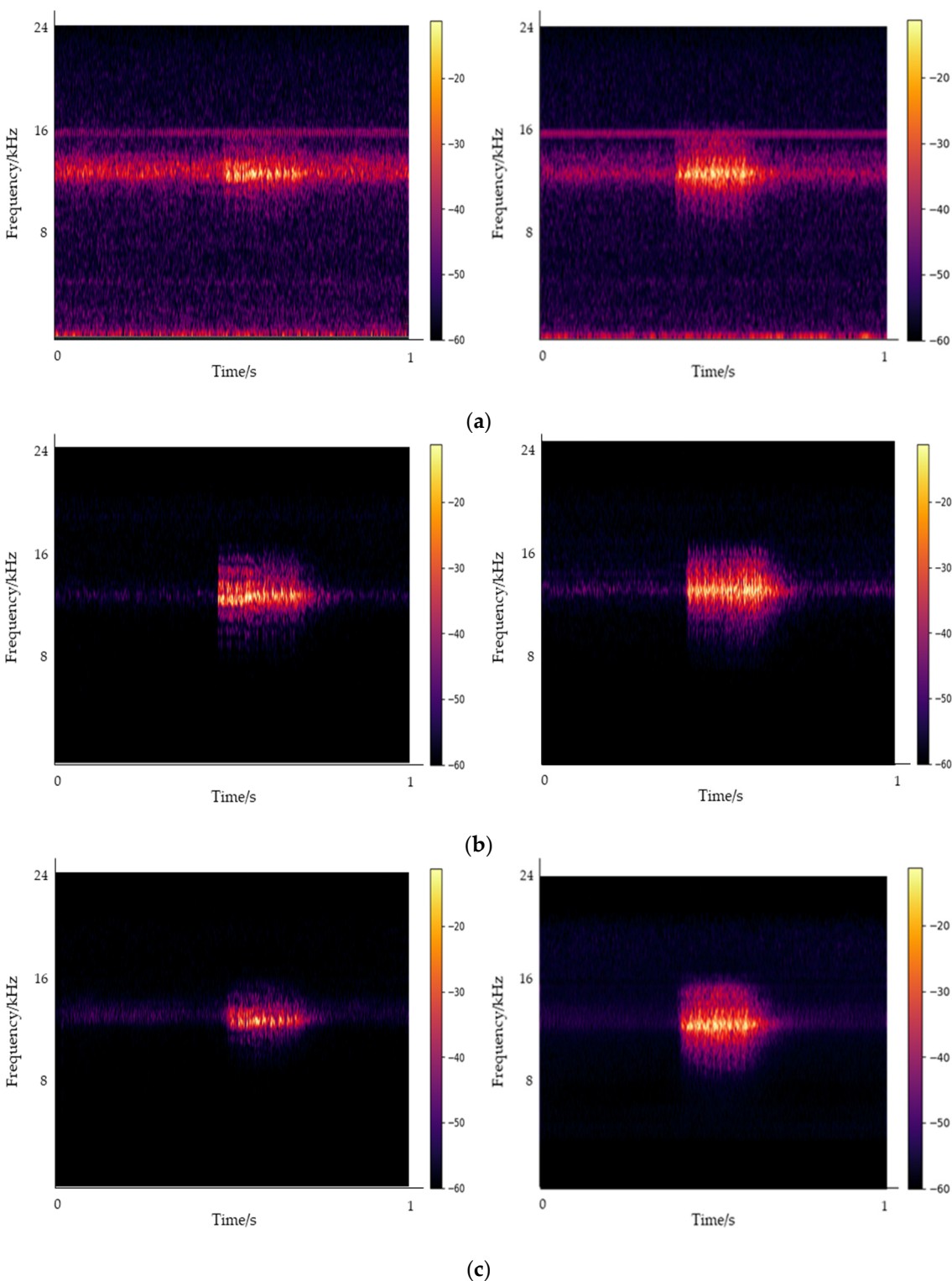

**Figure 11.** STFT analysis: (**a**) Noisy UUTs; (**b**) original signal; (**c**) estimated signal after the target noise reduction (TNR). SNRs of 0 and 10 dB are indicated on the left- and right-hand side panels, respectively.

## 4. Conclusions

This study focused on detecting electric sparks in the condensation environment of UUTs to prevent fires caused by electrical leakages that occur from condensation and corrosion owing to inadequate waterproofing, dehumidification, and ventilation facilities.

The results indicate that a microphone sensor can analyze the sound generated in UUTs and detect electric sparks, thereby contributing to the implementation of an intelligent fire detection system. The experimental results verify that the proposed method can stably detect electric spark sounds by eliminating the ventilation fan noise, which can have a significant impact on the UUTs. The proposed method can potentially address the limitations of traditional CCTV fire-detection technology. This can reduce the inefficiency of manual inspection and maintenance while increasing the efficiency of the system in managing UUTs. However, in the future, various pieces of equipment may be introduced to manage UUTs, and the diversity of sounds generated by such equipment can be a decisive factor in developing data-based anomaly detection. Therefore, further research is required to detect anomalies considering different types of sound data.

**Author Contributions:** Conceptualization, B.-J.L.; methodology, B.-J.L.; software, B.-J.L. and M.-S.L.; validation, B.-J.L. and W.-S.J.; writing—original draft preparation, B.-J.L.; writing—review and editing, B.-J.L. and W.-S.J. All authors have read and agreed to the published version of the manuscript.

**Funding:** This research was supported by the Institute of Information & Communications Technology Planning & Evaluation (IITP), grant funded by the Korea government (MSIT, MOIS, MOLIT, and MOTIE) (No. 2020-0-00061, Development of integrated platform technology for fire and disaster management in underground utility tunnel based on digital twin).

**Institutional Review Board Statement:** Not applicable.

**Informed Consent Statement:** Not applicable.

**Data Availability Statement:** Not applicable.

**Conflicts of Interest:** The authors declare no conflict of interest.

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
