# Peer review of "Acoustic Based Fire Event Detection System in Underground Utility Tunnels"

_fire, doi:10.3390/fire6050211_

Round 1

Reviewer 1 Report

My comments are shown below:

1)       Please check the spell of Equations. For example, in Equation 6, the word "otherwise" is unclear.

2)       In “Experimental Setup”, please introduce the basic information of the UUT selected in this study.

3)       More details on the hardware part of the detection system are needed.

4)       In section 2.3, “We searched open-source sound databases for sounds similar to electric sparks”. More discussions on the reliability and applicability of this are needed.

5)       The focus of this study is the detection of electric spark in UUT. However, the title of this study only mentions the prevention of fire, does not emphasize the detection of electric spark. It is suggested that modify the title appropriately and emphasize the key points.

No comments.

Reviewer 2 Report

In this manuscript, the author proposed an acoustic sensing-based anomaly detection system for prevention of fires in underground utility tunnels. The study is interesting and original. However, a more complete literature review should be carried out. The aim and the novelty should be strengthened. A more complete analysis of the results should be conducted.

There are a large number of editing errors in the text.

Reviewer 3 Report

Nothing  terribly exciting, but potentially useful

Minor proofreading needed

Author Response

1. Minor proofreading needed

- (MODIFIED) This paper has been re-edited with professional proofreading in English.

Round 2

Reviewer 2 Report

The authors respected all comments and made a major revision of the manuscript. They also substantiated all questions and comments. It is possible to give a manuscript for publishing.

English language is fine.